# Multi-Parameter Sensing in a Multimode Self-Interference Micro-Ring Resonator by Machine Learning

**DOI:** 10.3390/s20030709

**Published:** 2020-01-28

**Authors:** Dong Hu, Chang-ling Zou, Hongliang Ren, Jin Lu, Zichun Le, Yali Qin, Shunqin Guo, Chunhua Dong, Weisheng Hu

**Affiliations:** 1College of Information Engineering, Zhejiang University of Technology, Hangzhou 310023, China; 15868167064@163.com (D.H.); lujin@zjut.edu.cn (J.L.); qyl@zjut.edu.cn (Y.Q.); guosq@zjut.edu.cn (S.G.); 2Key Lab of Quantum Information, University of Science and Technology of China, Hefei 230026, China; clzou321@ustc.edu.cn (C.-l.Z.);; 3College of Science, Zhejiang University of Technology, Hangzhou 310023, China; lzc@zjut.edu.cn; 4State Key Laboratory of Advanced Optical Communication Systems and Networks, Shanghai Jiao Tong University, Shanghai 200240, China; wshu@sjtu.edu.cn

**Keywords:** self-interference micro-ring resonator (SIMRR), dissipative sensing, multimode sensing, multiparameter sensing, machine learning, artificial neuron network

## Abstract

A universal multi-parameter sensing scheme based on a self-interference micro-ring resonator (SIMRR) is proposed. Benefit from the special intensity sensing mechanism, the SIMRR allows multimode sensing in a wide range of wavelengths but immune from frequency noise. To process the multiple mode spectra that are dependent on multiple parameters, we adopt the machine learning algorithm instead of massive asymptotic solutions of resonators. Employing the proposed multi-mode sensing approach, a two-parameter SIMRR sensor is designed. Assuming that two gases have different wavelength dependence of refractive indices, the feasibility and effectiveness of the two-parameter sensing strategy are verified numerically. Moreover, the dependence of parameter estimation accuracy on the laser intensity noises is also investigated. The numerical results indicate that our scheme of multi-parameter sensing in a multimode SIMRR holds great potential for practical high-sensitive sensing platforms compared with the single-mode sensing based on whispering gallery mode (WGM) resonators.

## 1. Introduction

High-quality whispering gallery mode (WGM) optical resonators have emerged as a promising platform for compact and ultrahigh sensitive sensing over the past two decades [1,2,3], which have been studied theoretically and experimentally in various sensing applications, including solution and gas sensors, thermal sensors, humidity sensors, magnetometers [4,5,6,7,8,9,10,11,12]. Advantageously, WGM resonators can also detect, at a single particle level, nanoparticle, bio-molecule, and even atomic ions [13,14,15,16,17,18,19]. Based on WGM resonators, the sensing is typically realized by measuring the spectral changes in high resolution using a tunable laser or an optical spectrometer [6,7,8,9,10]. These spectral changes include mode shift, splitting and broadening, which is called reactive sensing [5,13,16,20,21,22,23,24]. Alternatively, when WGM resonators detect absorptive target analytes, the probe light may be strongly absorbed and then the linewidth of WGM resonator mode is significantly broadened, which is exploited to implement the dissipative sensing [16,20]. Lately, without the limitation of near-field interaction, a special intensity sensing has been presented in special waveguide-coupled WGM resonators [25,26,27,28]. In our previous works, the intensity sensing was exploited twice based on a microring resonator coupled with the sensing arm waveguide, which is called a self-interference microring resonator (SIMRR) [26,27,28]. The SIMRR sensor is designed only by exposing the target to be detected to the sensing arm or microring waveguide instead of the whole SIMRR structure. An induced tiny phase change of the sensing arm or microring waveguide mode is converted to a change of external effective coupling strength between the input waveguide and the microring waveguide, which then eventually leads to a significant change in the transmission depth, which is measured to realize the intensity sensing [26,27,28].

A remarkably neglected truth is that both reactive and dissipative sensing methods are always realized by precisely monitoring the spectral change of a single resonant mode or two coupled modes [5], and they are collectively referred to as single-mode sensing in the paper. Obviously, single- mode sensing based on WGM resonators is often not capable of multi-parameter detection [29,30,31,32,33]. For example, in practical environments, when a nanoparticle approaches the WGM resonator, the gap between the WGM resonator and input waveguide simultaneously becomes little, and single- mode sensing is unable to effectively identify two sensing targets. The reason for this is that the spectral changes of a single resonant mode for the abovementioned two situations are quite similar when two cases occur alone, and two sensing targets can not be distinguished from its spectral change. Therefore, the WGM sensor must address an open challenge to achieve the multi-parameter detection ability for practical applications [3,34,35,36]. Although a micro-ring resonator biosensing array is exploited to realize multi-parameter biosensing using wavelength division multiplexing technology [37,38,39,40,41,42], its complexarray structure and detecting cost hinder its further development. Hence, it is critical for WGM sensors to exploit a universal multi-parameter detection mechanism with low detection complexity. Recently, based on WGM resonators, a machine learning algorithm has been used for biological agents and micro/nano particles classification in [43,44]. A probabilistic neural network is used to classify the biological compounds with good accuracy by connecting with the resonant mode spectral frequency shift and the number of WGMs appearing within the free spectral range.

In this paper, SIMRR multimode sensing has firstly been proposed with the help of signal processing using an artificial neural network (ANN). The multimode sensing principle stems from the wavelength dependent spectral response of the SIMRR. When a tunable probe laser is used to excite the detecting systems, these transmission depths of multiple resonant modes can be collected for estimating the parameters. The back-propagation ANN (BP-ANN) is used for signal processing through training and test stages, where training and test data are constructed by these transmission depths within the wavelength range. As an example, a sensor of two gases using the SIMRR multimode sensing is numerically proved. The effect of the intensity noises on the multimode sensing performance are also studied, and the numerical results prove that the multimode sensing can achieve a lower limit of detection (LOD) at a high signal-to-noises ratio compared with that of single mode sensing. Multi-mode and multi-parameter sensing have great potential to offer a simple, robust and high-sensitive sensing platform with low detecting cost.

## 2. Multimode Sensing Mechanism by Self-Interference Micro-Ring Resonator 

Figure 1a shows the schematic diagram of the SIMRR proposed in [26], where the upper and lower coupling regions (denoted by the alphabets ‘U’ and ‘L’ in Figure 1a) are connected by a U-shaped sensing arm waveguide. As described in our previous works [27,28], a tiny phase change induced by the sensing arm waveguide or microring waveguide adjusts effectively the effective coupling strength between the waveguide and microring, and then makes a significant change in the resonant wavelength and transmission depth of a transmittance dip. Consequently, as the analyte is only exposed to the sensing arm waveguide or microring waveguide, the proposed single mode intensity sensing is realized by measuring the change of transmission depth in a single resonant mode. A simplified method using a transfer matrix method is applied to the derivation of the transmission coefficient T of the SIMRR, and the result is obtained as follows [26]:
(1)$$T=\frac{e^{-i\beta_{R}L_{R}/2}k-e^{-i\beta_{R}L_{W}}\left(1-k\right)+e^{-i\beta_{R}\left(L_{R}+L_{W}\right)}}{1-e^{-i\beta_{R}L_{R}}\left(1-k\right)+e^{-i\beta_{R}\left(L_{R}/2+L_{W}\right)}k}$$
where β_R_ is the propagation constant of the waveguide mode (for simplicity we assume that all the waveguides have the same propagation constant), L_R_ denotes the physical length of the microring, L_W_ denotes the effective length of the sensing arm waveguide, and k is the power coupling coefficient of two symmetric directional coupling regions. Concretely, the propagation constant of the waveguide mode equals β_R_ = (2π/λ)n_eff_ − iα, where λ is the optical wavelength, n_eff_ is the effective refractive index of the waveguide and α is the loss coefficient per unit length of the optical waveguide mode. The sensor can be fabricated in Silicon-on-Insulator (SOI) technology with a standard SOI wafer, and the detailed fabrication procedure can be found in [10,31].

In the designed sensor in Figure 1a, the analyte to be detected is only exposed to the sensing arm waveguide. The effective length of the sensing arm waveguide is described by L_W_ = L_L_ + l, where L_L_ represents its initial physical length when no analyte is exposed to it, and l represents the induced length change of the sensing arm waveguide. In essence, the length change l is equivalent to the refractive index change of sensing arm waveguide. In the designed SIMRR sensor, a microring was chosen with a radius of R = 30 µm and a 220 nm high and 500 nm width ridge waveguide. The effective refractive index of its fundamental mode equals n_eff_ = 2.45, and the waveguide loss coefficient is set to α = 0.1 dB/cm. The initial physical length of sensing arm waveguide is set to L_L_ = 250 µm, and the power coupling coefficient of the directional coupling region is set to k = 0.5. Figure 1b displays the simulated transmission spectra in the wavelength range from 1400 to 1600 nm for the cases of l = −10 nm, 0 and 10 nm (here, the large step size is chosen in order that the spectral changes are easy to be identified in Figure 1b,c). Its typical spectrum is remarkably different from that of conventional coupled microring waveguide system. In the latter, the peaks or dips show a periodical oscillation in wavelength, while in the former the transmittance dips with different transmission depths appear irregularly in wavelength. When the length change l of the sensing arm waveguide is adjusted, the SIMRR exhibits completely different wavelength dependent responses. Figure 1c is the enlarged diagram of Figure 1b in the wavelength range from 1442 nm to 1449 nm, where two transmittance dips are present near 1443 nm and 1448 nm at a fixed value of l. With the increasing of the induced length change l, the resonant wavelengths of transmittance dips are slightly shifted, and simultaneously their transmission depths are significantly varied, where the latter has been exploited to realize intensity sensing in a single resonant mode [26,27,28]. However, it is observed that their transmission depths do not show a monotonous change with l, and the single mode intensity sensing cannot be realized within the range of −10 nm ≤ l ≤ 10 nm. With the increasing length change l, the transmission depth near 1443 nm gets smaller and then bigger. Conversely, the transmission depth near 1448 nm gets bigger and then smaller. Obviously, the transmission depths near 1443 nm and 1448 nm have an opposite variation trend with l. In nature, all the transmission depths within the wavelength range of 1400 nm ≤ λ ≤ 1600 nm can be adopted as the effective sensing information, and then it is critical to fusion these effective sensing information to improve the sensitivity. Consequently, within a broad wavelength range, these transmission depths of multiple resonant modes can be fully used to realize multi-mode sensing.

The conventional probe laser scanning (the method is shown in [27]) can be used to collect these transmission depths of the multiple resonant modes. When the SIMRR sensor is excited by a tunable laser via an optical tapered fiber, a photodiode is connected with the output port of the SIMRR via an optical tapered fiber, and an oscilloscope is ultimately used to collect its transmission spectra. These transmission depths of the multiple resonant modes can be easily extracted from the obtained transmission spectra. Subsequently, a machine learning technique will be used for multimode sensing to efficiently fusion these effective sensing information.

## 3. Multimode Sensing Based on Artificial Neuron Network

These collected effective sensing signals (transmission depths) from the designed SIMRR sensor are processed based on a BP-ANN. The ANN is widely known machine learning techniques, which has been developed to tackle a variety of problems in many sensing areas [45,46,47,48,49]. As shown in Figure 2a, the BP-ANN consists of three types of layers, i.e. the input layer, hidden layer and output layer, which are connected by lines known as weights or links. When the designed sensor is used in Figure 1a, a length change l of sensing arm waveguide is induced by the target to be detected, whose spectra have a group of transmission depths. Hence, each induced length change l is related with one group of the transmission depths, which are adopted as the input variables (xi) and applied to I input nodes. These variables are multiplied by a value of weight factor w_im_ (*i* = 1, … *L*; *m* = 1, … *M*) corresponding to the input layer of links. Here M is the number of nodes in the hidden layer, and it has an important impact on the learning ability of the BP-ANN. Similarly the signals in the hidden layer are multiplied by another weight factor w_mn_ (*n* = 1, …, *N*), *N* being the node number of output layer. The output layer is used to estimate the sensing parameter depending on the corresponding input variables. For the proposed multimode sensing in SIMRR, without loss of generality, the single output parameter (N = 1) is selected as the length change l of the sensing arm waveguide, because the quantity can represent the object to be detected in the general sense. Then the final output variable can be expressed as,
(2)$$\hat{l_{t}}=\overset{M}{\underset{m=1}{\sum }}w_{m}f\left(\overset{I}{\underset{i=1}{\sum }}w_{im}x_{i}\right)$$
where f(·) is generally chosen to be a sigmoid function. There are two steps when carrying out signal processing using the BP-ANN. In the first step, a training set is constructed by many pairs of input variables and the corresponding theoretical output, and they are generated theoretically or experimentally in order to train the network. The purpose of training network is to minimize the mean squared error (MSE) E_e_, which is often defined as the average sum of the squares of the errors between the output estimated value $\hat{l_{t}}$ of the length change and the corresponding theoretical output l_t_,
(3)$$E_{e}=\frac{1}{T}\overset{T}{\underset{t=1}{\sum }}{\left(l_{t}-\hat{l_{t}}\right)}^{2}\;\;\;\;\;\;\;\;\;\;\;\;$$
where T represents the total number of measurements, and t represents an index of measurements. By means of the gradient-descent-based error back-propagation algorithm, these weight factors are interactively changed to minimize the error function E_e_ as:(4)$$\Delta w_{ji}^{\tau }=w_{ji}^{\tau }-w_{ji}^{\tau -1}=-\eta \frac{\partial E_{e}}{\partial w_{ji}}|w^{\tau }$$
where η is the learning rate. The derivative of E_e_ is assessed by the derivative algorithm known as backpropagation, and a detailed account of the backpropagation procedure can be found in [48]. After the best training performance is achieved, the trained network has realized the expectation aim and these weight factors are saved. When the object to be detected is detected by the SIMRR sensor its spectra can be collected by the probe laser scanning method, and a group of transmission depths within the wavelength range is extracted from the obtained spectra as a test set. In the second step, as the input variables, the test set is input into the trained neural network, and the object to be detected can be estimated by its corresponding output parameter based on empirical knowledge gained through the training process.

In the following, the feasibility and effectiveness of the multi-mode sensing strategy are verified by the numerical simulations. In this work, the length change of the sensing arm waveguide is set within the range of −0.01 µm ≤ l ≤ 0.01 µm and their corresponding transmission spectra can be calculated according to Equation (1). Within the wavelength range of 1400 nm ≤ λ ≤ 1600 nm, there are 28 transmittance dips present, and then the number I of nodes in the input layer is set to 28. Other initial neural network parameters are set as follows: The node number of 10 in the hidden layer, the learning rate of 0.01, the training goal error of 1 × 10^−6^, and the number of units of 100 for training dataset. The initial training dataset is constructed by many pairs of the theoretical value of the length change l and its corresponding transmission depths, which are obtained from the spectra calculated by changing the length change l from −9.999 nm to 9.999 nm with the step size of 0.202 nm. As shown in Figure 2b–e, the effect of network parameters on its measurement results, such as number of neurons in the hidden layer, learning rate, number of units for training dataset and training goal error, are studied in order to optimize the network performance. When optimizing a parameter, the MSE E_e _ versus the parameter is studied with other given parameters, and then the current parameter gets the optimal value on the minimum value of the MSE E_e_. Subsequently, the next parameter is optimized continuously after replacing the current parameter with its optimal value in the given parameter set. The final optimized parameter values are as follows: The node number of 15 in the hidden layer, the learning rate of 1 × 10^−5^, the training goal error of 1 × 10^−8^, and the number of units of 900 for training dataset. The number of units for training dataset can be increased by decreasing the step size of the length change l, and the number of units of 900 for training dataset is obtained by changing the length change l within the range of −9.99688 nm ≤ l ≤ 9.99688 nm with the step size of 0.02224 nm. The test dataset consists of many groups of transmission depths, and they are obtained from these spectra, which are calculated by substituting the theoretical value of length change l into Equation (1). In Figure 2f, the theoretical values of length change are set within the range of -0.01 µm ≤ l ≤ 0.01 µm with the step of 0.4 nm, 0.5 nm and 0.8 nm, respectively, and their estimated average values of length change after 500 simulations separately make up of three lines from right to left. The estimated values are compared with the corresponding theoretical values of length change l. It is clear that the estimated results do agree well with the theoretical values, where their total MSE achieves the value of 1.56 × 10^−13^. The simulation results prove that the length change l of sensing arm waveguide can be measured exactly by using the multi-mode sensing method.

In summary, in combination with signal processing using BP-ANN, the SIMRR multi-mode sensing has been proposed to improve detection accuracy. The multi-mode sensing mechanism stems from the special transmission spectra of the SIMRR, which exhibit different wavelength responses in multiple resonant modes. The BP-ANN for signal processing is required to be trained in advance, and the training datasets are constructed by many groups of transmission depths within the wavelength range paired with the corresponding known output values. When the object to be detected is detected by the designed sensor, these corresponding transmission depths are obtained and are input into the trained BP-ANN, and the network output is the estimated value of the unknown object.

## 4. Multi-Parameter Sensing by Self-Interference Micro-Ring Resonator

It is often necessary for WGM resonators to be capable of the simultaneous sensing of multiple elements in practical applications. As previously mentioned, although a microring resonator array has been exploited to realize the multi-parameter sensing with the aid of wavelength multiplexing technology, the universal multi-parameter sensing method has high detection complexity. Lately, a novel approach to simultaneous measurement of ammonia vapor and humidity in air has been demonstrated by exciting WGMs in an array of two microspheres [41]. Two microspheres are coated with different polymers, and they separately have a dominant sensitivity over ammonia and humidity. Ammonia concentration and relative humidity can be distinguished by separately monitoring the special spectral positions of two WGM resonances relevant to each of the microspheres. However, the method does not work if the special spectral positions relevant to each of the microspheres are not present. In this section, based on the SIMRR multimode sensing in Section 2 and Section 3, a universal multi-parameter sensing approach is proposed with reduced detection complexity.

### 4.1. The Designed Two-Parameter Sensor by Self-Interference Micro-Ring Resonator

A two-parameter SIMRR sensor is designed in Figure 3a. Here, in order to detect two kinds of gases, the surfaces of two half-ring waveguides in the left and right are separately coated by two gas sensitive materials. The cladding layer refractive index shift ∆n_c_ is caused due to the interaction between gas molecules and the corresponding sensitive materials, and then it is related to the wavelength (λ) and gas molar concentration (C). The refractive index shift further leads to an effective index change ∆n_eff_ of mode propagating around the micro-ring waveguide, which can be described as,
(5)$$\Delta n_{eff}\left(\lambda ,C\right)=S_{w}\Delta n_{c}\left(\lambda ,C\right)$$
where S_w_ is the dielectric waveguide sensitivity. The waveguide sensitivity depends on the geometry and material of the waveguide, which is obtained as follows:(6)$$\Delta n_{c}\left(\lambda ,C\right)=F\left(\lambda ,\lambda_{0}\right)\varepsilon \left(\lambda_{0}\right)\nu_{t}S_{b}pC$$
where *λ_0_* is the central wavelength in absorption band, F(*λ, λ_0_*) is the proportionality factor betweenabsorption and effective index changes, ν_t_ is the total number of BCP (bromocresol purple) molecules per PMMA (polymethylmethacrylate) unit of volume, S_b_ is the reagent-analyte binding constant, and p is the polymer permeability factor. Due to two different interactions between two kinds of gas molecules and their corresponding sensitive materials, the effective index changes ∆n_eff _ of two half-ring waveguides in the left and right exhibit the different responses in multiple resonant modes. As is known to all, almost all WGM-based sensors have focused on the measurement of a single mode, which results in the throw away of experimental data in other modes of different wavelengths. In order to prove two-parameter sensor’s feasible numerically, it is assumed that their effective refractive index changes exhibit two different functions of wavelength. For simplicity, the effective refractive index changes ∆n_eff_ of two half-ring waveguides are the Gaussian and Lorentz functions of wavelength, respectively, which are given as followed,
(7)$$S_{w}=\frac{dn_{eff}\left(\lambda \right)}{dn_{c}\left(\lambda \right)}$$

Assuming that one of two gases is ammonia, and the polymethylmethacrylate (PMMA) doped with bromocresol purple (BCP) has been selected as sensitive material in the cladding layer to detect ammonia [50]. As PMMA-BCP absorption band extends from 350 nm to 450 nm, a cladding layer refractive index shift ∆n_c_ at wavelengths around 1550 nm is given by,
(8)$$\Delta n_{eff1}\left(\lambda ,C_{1}\right)=0.02975C_{1}\exp\left[\frac{-{\left(\lambda -1550\right)}^{2}}{5000}\right]\;\;$$
(9)$$\Delta n_{eff2}=0.02975C_{2}\frac{100}{{\left(\lambda -1550\right)}^{2}+100}$$
where C_1_ and C_2_ denote separately two gas molar concentrations. In order that the refractive index changes are in line with the reality, the parameters in Equation (8) and Equation (9) are elaborately selected so that their effective index changes are close to the corresponding actual values in [50]. The effective refractive index of two half-ring waveguides in the left and right can be separately given by n_eff1_ = n_eff_ + ∆n_eff1_ and n_eff2_ = n_eff_ + ∆n_eff2_, and the length of the unexposed sensing arm waveguide equals L_W_ = L_L_ = 250.005 µm. The other structural and physical parameters are the same with that in Figure 1a. Figure 3b displays their effective refractive indices versus wavelength at C_1_ = C_2_ = 5%. It can be seen that their refractive index curves are quite different from each other, whose values are essentially in agreement with the actual values [50].

### 4.2. Simulation Verification Based on Artificial Neuron Network

In this section, the two-parameter SIMRR sensing is verified by numerical simulation. According to Equation (1), the two-parameter sensor’s transmission coefficient T can be rewritten into,
(10)$$T=\frac{e^{-i\beta_{R1}L_{R}/2}k-e^{-i\beta_{R}L_{L}}\left(1-k\right)+e^{-i\left(\beta_{R1}L_{R}/2+\beta_{R2}L_{R}/2+\beta_{R}L_{L}\right)}}{1-e^{-i\left(\beta_{R1}L_{R}/2+\beta_{R2}L_{R}/2\right)}\left(1-k\right)+e^{-i\left(\beta_{R2}L_{R}/2+\beta_{R}L_{L}\right)}k}$$
where the propagation constants of two half-ring waveguides in the left and right separately equal β_R1_ = (2π/λ)(n_eff_ + ∆n_eff1_) ) − iα and β_R2_ = (2π/λ)(n_eff_ + ∆n_eff2_) − iα. Then, the sensor’s spectra can be calculated numerically according to Equation (10), from which the required training and test sets can be extracted. Subsequently, the two-parameter sensing can be realized with the help of signal processing using a BP-ANN. As shown in Figure 4a, a two-output BP-ANN is designed for signal processing of the two-parameter sensor. The input variables come from the transmission depths in the calculated transmission spectra, x_i_, and two output parameters (N = 2) correspond to two gas concentrations. Each group of training dataset is constructed by one group of transmission depths within the wavelength range paired with two known gas concentrations C_1_ and C_2_. With two output parameters being adopted, a huge amount of training data is required in the BP-ANN. Therefore, several small concentration ranges are separately adopted for the training and test set selection, including 1% ≤ C_1_(C_2_) ≤ 0.2 and 0.05% ≤ C_1_(C_2_) ≤ 0.1%. The input transmission depths are obtained from the spectra, which are calculated by substituting two gas concentrations into Equations (8)–(10). As two gas concentrations are set to a molar concentration of 1% or 5%, the transmittance dips within the wavelength range of 1400 nm < λ < 1600 nm are displayed in Figure 4b. These transmission depths are marked by the labels of square and circle for the gas molar concentrations of 1% and 5%, respectively. With two gas concentrations being changed, these transmission depths show irregular changes in multiple resonant modes.

The network parameters of the BP-ANN are optimized in the same way that they are optimized for the BP-ANN in Figure 2a. When 1% ≤ C_1_(C_2_) ≤ 0.2, the obtained optimized parameters are as follows: The node number of 21 in the input layer, the node number of 15 in the hidden layer, the learning rate of 0.01, and the training goal error of 5 × 10^−5^. In addition, we have investigated the effect of wavelength range and threshold of transmission depth on the measurement results. It is necessary to determine how much wavelength range is suitable to collect the transmission depths for the BP-ANN. In the example, the wavelength range always centered at 1550 nm, and could be increased or decreased to investigate the performance of the BP-ANN. Figure 4c displays the MSE versus the wavelength range curve. Noted that the MSE can achieve a lower value when the wavelength range is greater than 150 nm, and the 150 nm is enough to monitor accurately two gas concentrations using the ANN in the case. Thus, the wavelength range from 1400 nm to 1600 nm is adopted in the remainder of the paper. Within the wavelength range, only the transmission depths are selected as the input variables of the BP-ANN when they are less than the value, which is defined as the threshold value of transmission depth. When the threshold value is set range at 0.75 to 1, Figure 4d displays MSE versus the threshold value of transmission depth curve. The MSE achieves the lower values when the threshold is located between 0.8 and 0.96. The reason can be explained as followed. When the threshold value is smaller than 0.8, the number of the transmission depths is very little within the wavelength range so that the approach loses a great deal of useful sensing information. While the threshold value is larger than 0.96, there are so many transmission depth values close to 1 present within the wavelength range. However, they show a slight change with two gas concentrations being changed (which can be seen from Figure 4b). Hence, these transmission depths is ineffective for the sensing, and they may unfavorably affect the accuracy of measurement results by the BP-ANN. The threshold value is chosen as 0.8 in such a case. In order to effectively reduce training set, the set is chosen as two gas concentrations that are equal to a certain value C_0_ in this paper. When the value is changed from a molar concentration of 1% to 0.2 with the step size of 0.0019, each group of training data is constructed by one group of transmission depths within the wavelength range paired with two known gas concentrations of C_1_ and C_2_. At 1% ≤ C_1_(C_2_) ≤ 0.2, the test set is selected to investigate the performance of the BP-ANN when the gas concentration C_1_ is adjusted within the range of 0.01187 ≤ C_1_ ≤ 0.19983 with the step of 0.04699 and the gas concentration C_2_ is adjusted within the range of 0.01819 ≤ C_2_ ≤ 0.18277 with the step of 0.02743. The output measurement results by the BP-ANN are compared with the theoretical values of two gas concentrations, and they are in accord with these theoretical values. However, the concentration range of a molar concentration of 1% to 0.2 is not realistic, so a small concentration range of 0.05% ≤ C_1_(C_2_) ≤ 0.1% is used to investigate the performance of the method. In such a case, the obtained optimized network parameters are as follows: The node number of 16 in the input layer, the node number of 8 in the hidden layer, the learning rate of 0.01, and the training goal error of 5 × 10^−5^. The threshold value of transmission depth is optimized as 0.9. The training set is chosen when C_0_ is changed from a molar concentration of 0.05% to 0.1% with the step size of 0.0005%. The test set is selected when the gas concentration C_1_ is adjusted within the range of 0.050138% ≤ C_1_ ≤ 0.099848% with the step of 0.012426% and the gas concentration C_2_ is adjusted within the range of 0.052049% ≤ C_2_ ≤ 0.094905% with the step of 0.0071426%. Obviously, there are five values of gas concentration C1 and seven values of gas concentration C_2_ within a small concentration range of 0.05% ≤ C_1_(C_2_) ≤ 0.1%. Thus the test set corresponds to 35 values of two gas concentrations C_1 _ and C_2_, which are considered as the test indices. The selection of test set almost cover the small concentration range, and the measurement does not lose its generality. As shown in Figure 4(e) and (f), the measurement results by the BP-ANN are compared with the theoretical values of two gas concentrations, and they do agree well with these theoretical values. The total number of measurements is set to 500, and their MSEs achieve the value of 9.4132 × 10^−7^. The numerical results have demonstrated that two gas concentrations can be measured accurately in the designed two-parameter sensor.

In the same way, the method can be easily expanded to a multi-parameter ( the number of parameter N_para_ > 2) sensing in SIMRR, and the proposed method offers a universal multi-parameter sensing approach with low detection complexity.

## 5. Multimode and Multi-Parameter Sensing under Intensity Noises

All of the above studies are based on ideal model without considering the effect of the noises. In practical applications, experimental noises are mainly categorized into the frequency and intensity noises [28]. The former results in the uncertainty of the variation in the resonant wavelength of transmission dip, and the latter leads to the variation in the depth of transmission dip [32,33,51]. In our previous research, it was theoretically and numerically proved that the variation in the transmission depth was almost immune from the frequency noises at a high signal-to-noise ratio (SNR) [28]. While in the proposed multimode sensing, only the transmission depths are utilized without considering their resonant wavelengths, and the effect of the frequency noises on the transmission depths can be effectively removed by expanding the wavelength range [28]. Thus, only the effect of intensity noises on the approach is investigated in this work.

For the designed SIMRR sensor in Figure 1a, Figure 5a shows the simulated transmission spectra in the wavelength range of 1400–1600 nm at l = −10 nm without noises. The transmission spectra under various noise levels are obtained by adding the small Gaussian noise with zero mean to a noise-free spectra. A given SNR value can be obtained by adjusting the variance of Gaussian noise. Figure 5b,c display the transmission spectra at the SNR values of 15 dB and 30 dB, respectively. Noted that the intensity noises have great influences on the transmission depths at SNR = 15 dB, while they have less influence on the transmission depths at SNR = 30 dB and these depths almost do not change compared with the corresponding values without noises in Figure 5a. At a given SNR value, the training and test sets are constructed separately by the collected transmission depths with noises, and they are input to the BP-ANN to investigate the influence of intensity noises on measurement results. In the presence of intensity noises, Figure 5d displays root-mean-square error (RMSE) of estimated parameter versus SNR. The RMSE under the intensity noises descends with the increasing of the SNR value, and keeps a stable value of about 10^−6^ nm at SNR > 48 dB, which is in accord with the RMSE level under noise free condition. In this work, after the measurement is repeated for 500 times, for the multimode sensing scheme, its RMSE is approximately equal to its limit of detection (LOD). As a comparison, the single mode sensing is also used by monitoring the variation in the transmission depth of a single resonant mode, and the detailed derivation of its LOD is provided by Equation (A1) in Appendix A. Figure 5d also shows the LOD of the single mode sensing method as a function of the SNR value. At 30 dB > SNR > 48 dB, the RMSE level of the multimode sensing is more than two orders of magnitude lower than the LOD of the single mode sensing. When SNR > 48 dB, the RMSE level of the multimode sensing is gradually close to the LOD of the single sensing. These numerical results prove that the multi-mode sensing under intensity noises have a significant improvement in the accuracy of detection compared with the single mode sensing. Obviously, the multi-mode sensing gets more effective sensing information than the single mode intensity sensing, and it has lower LOD than the single mode intensity sensing.

For the designed two-parameter SIMRR sensor in Figure 3a, Figure 6a shows the simulated transmission spectra in the wavelength range of 1400–1600 nm at C_1_ = C_2_ = 0.01 without noises. Figure 6b,c show the corresponding spectra under laser intensity noises at the SNR values of 15 dB and 30 dB, respectively. It is observed that the intensity noises have great impact on the transmission depths at SNR = 15 dB, while the transmission depths at SNR = 30 dB are immune from the intensity noises. At 0.05% ≤ C_1_(C_2_) ≤ 0.1%, under various noise levels, the two-parameter SIMRR sensor is used, and the performance of the BP-ANN in Figure 4a is investigated. Figure 6d displays the MSE of estimated concentrations versus SNR curves. With the increasing of the SNR value, the mean square errors (MSEs) of C_1_ and C_2_ decrease. At SNR = 60 dB, the MSEs of C_1_ and C_2_ are both close to their corresponding values without noises. As the single mode intensity sensing method can not realize the two-parameter detection, no results by the method are submitted to compare with the MSEs by the multi-mode sensing method. These numerical results prove that the proposed multi-mode sensing can carry out highly accurate two-parameter sensing at high SNR.

## 6. Conclusions

A systematical investigation on the performance of the multi-mode SIMRR sensor was presented in this paper. The multimode sensing mechanism results from the different responses to the analyte to be detected in multiple resonant modes, i.e. the different transmission depth changes in multiple resonant wavelengths. After the transmission spectra were obtained by scanning the probe laser wavelength, the transmission depths within a wavelength range are extracted as effective sensing features and input into a BP-ANN for signal processing. The multi-mode sensing was exploited for the multi-parameter detection, and a two-parameter detection was numerically demonstrated based on the designed SIMRR two-parameter sensor. The dependence of the multi-mode sensing performance on the laser intensity noises was also studied. Compared with the single mode sensing, the multi-mode SIMRR sensing had the potential to achieve a much lower LOD at a high SNR.

## Figures and Tables

**Figure 1 sensors-20-00709-f001:**
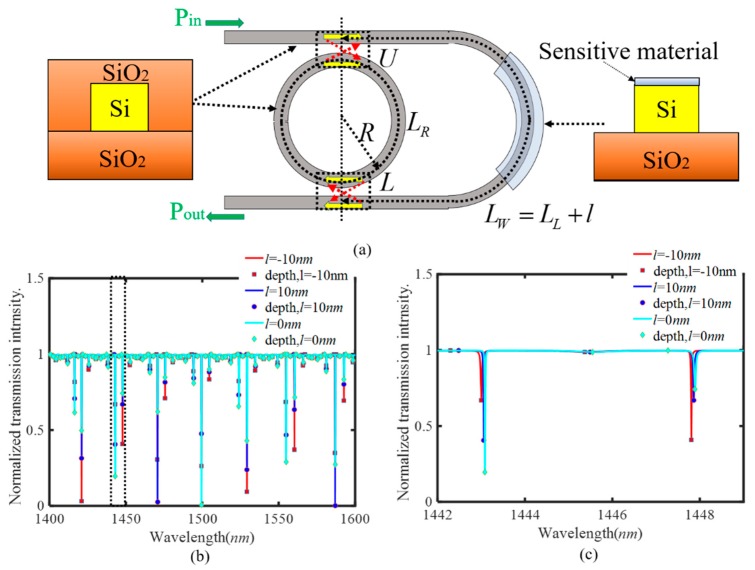
Multimode sensing mechanism by SIMRR (self-interference micro-ring resonator). (**a**) Structural diagram of SIMRR and its sensor design. (**b**) The simulated transmission spectra in the wavelength range of 1400–1600 nm for the cases of the length change of sensing arm waveguide l = −10, 0, 10 nm. (**c**) Enlarged diagram of dotted area in (**b**).

**Figure 2 sensors-20-00709-f002:**
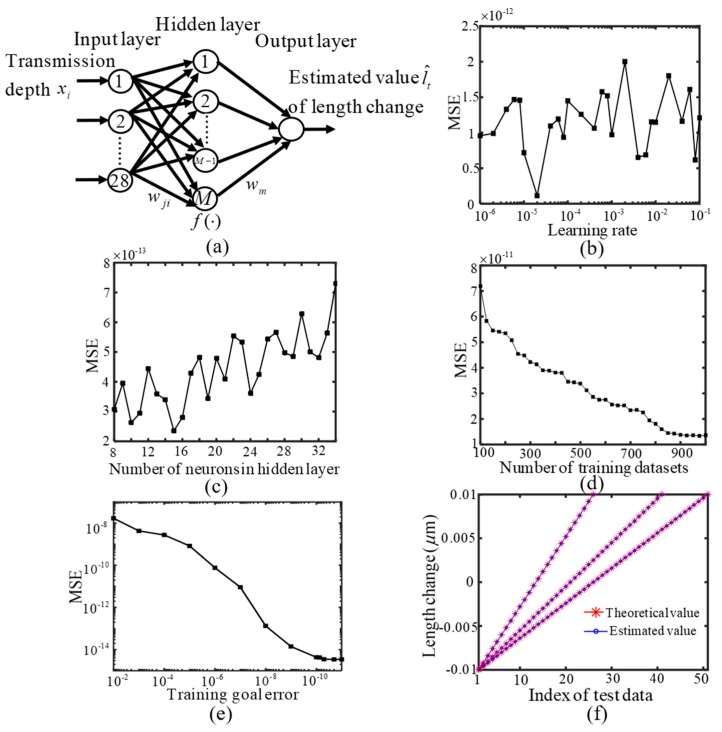
SIMRR multi-mode sensing based on artificial neuron network. (**a**) Signal processing model using BP-ANN (back-propagation ANN) for the designed SIMRR multi-mode sensor in the Figure 1a. Influence of the network parameters on its measurement results, (**b**) MSE (mean squared error) versus number of neurons in the hidden layer, (**c**) MSE versus learning rate, (**d**) MSE versus number of training datasets, (**e**) MSE versus training goal error, (**f**) Comparison between the estimated values of the length change and their corresponding theoretical values.

**Figure 3 sensors-20-00709-f003:**
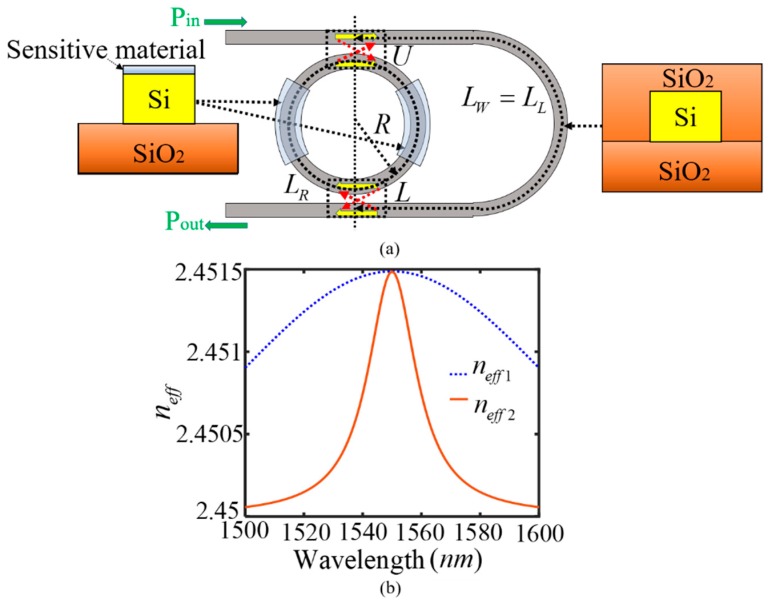
The designed two-parameter SIMRR sensor. (**a**) Structural diagram of the designed two-parameter SIMRR sensor. The surfaces of two half-ring waveguides in the left and right are separately coated by two sensitive materials to detect two kinds of gases. (**b**) when two gases to be detected are only exposed to two half-ring waveguides in the left and right, it is assumed that their effective refractive index changes are the Gaussian and Lorentz functions of of wavelength, respectively. At C1 = C2 = 5%, their waveguide effective refractive indices can be separately given by n_eff1_ = n_eff_ + ∆n_eff1_ and n_eff2_ = n_eff_ + ∆n_eff2_.

**Figure 4 sensors-20-00709-f004:**
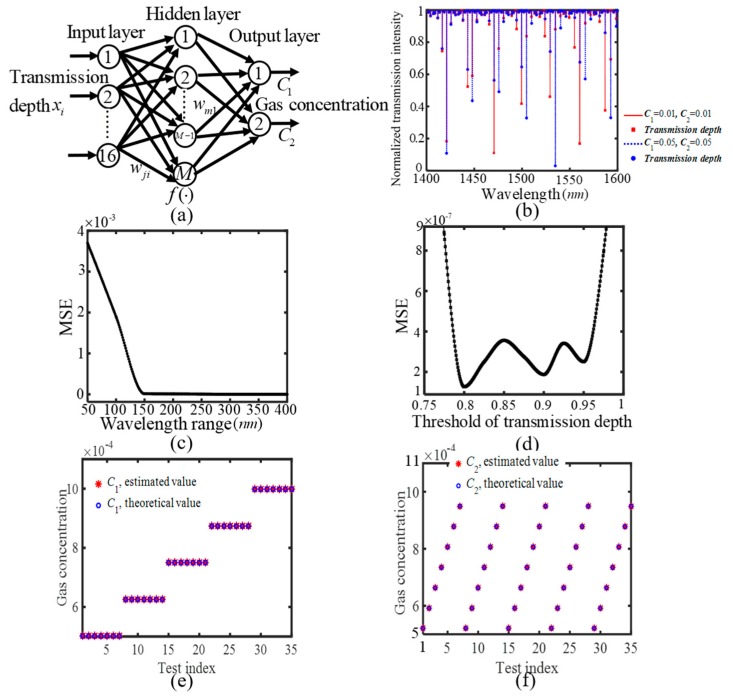
Multi-parameter SIMRR sensing using BP-ANN. (**a**) a BP-ANN for signal processing of the two-parameter SIMRR sensor in Figure 3a. (**b**) The simulated transmission spectra in the wavelength range of 1400–1600 nm at C_1_ = C_2_ = 1% or 5%. (**c**) MSE versus wavelength range, (**d**) MSE versus threshold value of transmission depth, (**e**) Comparison between the estimated values by BP-ANN and the theoretical value of gas concentration C_1_, (**f**) Comparison between the estimated values by the BP-ANN and the theoretical value of gas concentration C_2_.

**Figure 5 sensors-20-00709-f005:**
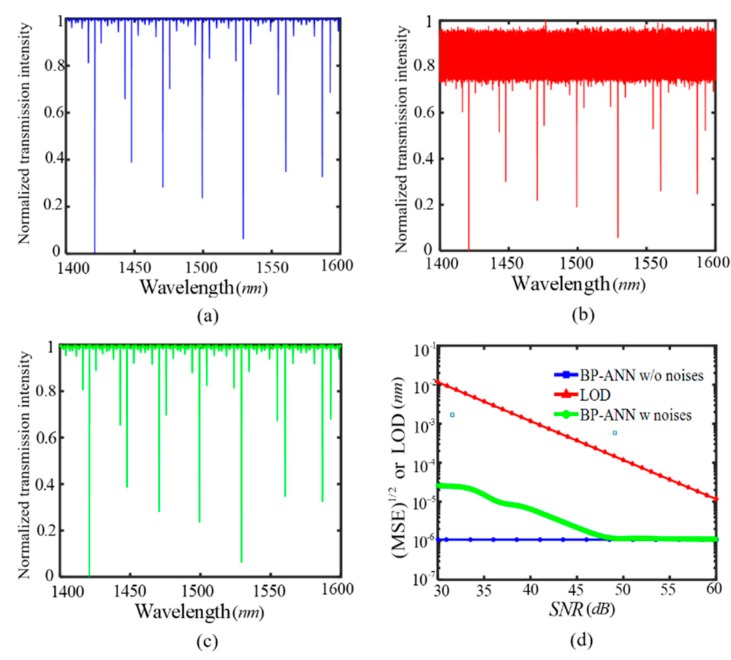
For the designed SIMRR sensor in Figure 1 (**a**), the simulated transmission spectra in the wavelength range of 1400–1600 nm at the length change of sensing arm waveguide l = −10 nm (**a**) without noises, (**b**) with SNR = 15 dB, (**c**) with SNR = 30 dB. (**d**) The root-mean-square error (RMSE) of the measurement results by the BP-ANN and the LOD (limit of detection) of single mode dissipative sensing versus the SNR value.

**Figure 6 sensors-20-00709-f006:**
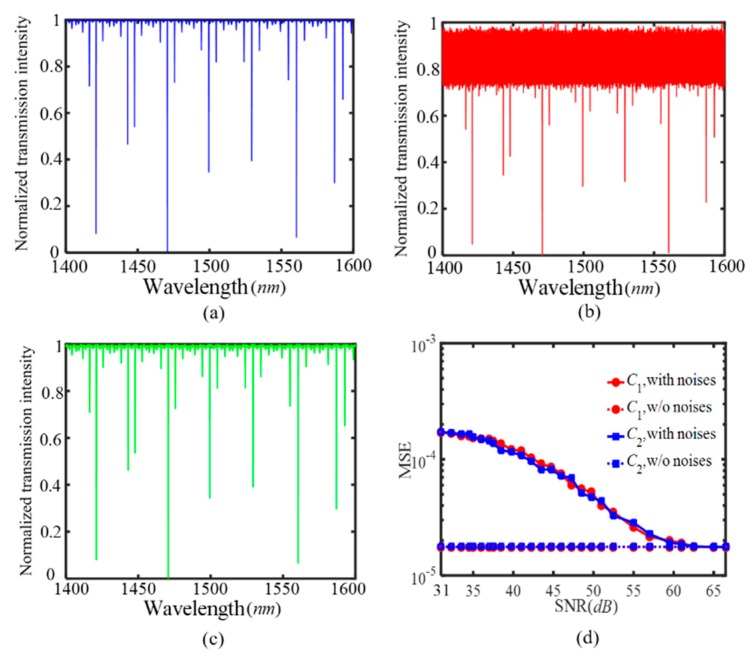
For the designed two-parameter SIMRR sensor in Figure 3a, the simulated transmission spectra in the wavelength range of 1400–1600 nm at gas concentration C_1_ = C_2_ = 0.01 (**a**) without noises, (**b**) with SNR = 15 dB (**c**) with SNR = 30 dB. (**d**) The root-mean-square error of measurement results by the BP-ANN versus the SNR value.

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
