# Peer review of "Multi-Parameter Sensing in a Multimode Self-Interference Micro-Ring Resonator by Machine Learning"

_sensors, 2020, doi:10.3390/s20030709_

Round 1

Reviewer 1 Report

The changes and corrections the authors have made to this paper have significantly improved it. I have a few final comments:

I still think this is better described as a dispersive technique, because the critical phenomenon as modeled is a change in the dispersion of the sensing arm or ring. I understand that this also changes the amount of dissipation in the ring, but I feel the paper may be interesting to more people if the role of dispersion is emphasized more strongly.

I picked up one more ambiguity: lines 79 and 87 describes L_W as the physical length of the sensing arm, but L_L is the physical length, and l is the effective change in length. L_W should be called the effective length, or similar. This should definitely be clarified, this is quite a confusing sentence as it stands.

Author Response

Reviewer 1

The changes and corrections the authors have made to this paper have significantly improved it. I have a few final comments:

I still think this is better described as a dispersive technique, because the critical phenomenon as modeled is a change in the dispersion of the sensing arm or ring. I understand that this also changes the amount of dissipation in the ring, but I feel the paper may be interesting to more people if the role of dispersion is emphasized more strongly.

I picked up one more ambiguity: lines 79 and 87 describes L_W as the physical length of the sensing arm, but L_L is the physical length, and l is the effective change in length. L_W should be called the effective length, or similar. This should definitely be clarified, this is quite a confusing sentence as it stands.

Reply: First, we would like to thank the Referee for carefully reading of our revised manuscript and raising valuable remarks.

The Referee still thinks the proposed single mode dissipative sensing should be better described as a dispersive technique, although he (or she) understands that it also changes the amount of dissipation in the microring. According to this, in the revised manuscript, we replace “dissipative sensing” by “special intensity sensing”, which has ever been called in the reference (H. Ren, C.-L. Zou, J. Lu, L. Xue, S. Guo, Y. Qin, W. Hu, “Highly-sensitive intensity detection by a self-interference micro-ring resonator,” IEEE Photon. Tech. lett., 2016,28, pp.1469- 1472.).

In the lines 79 of the revised paper, the sentence is rewritten as “……LR denotes the physical length of the microring, LW denotes the effective length of the sensing arm waveguide,……”. In the lines 86 of the revised paper, the sentence is rewritten into“…… LL represents its initial physical length when no analyte is exposed to it, ……”.

Reviewer 2 Report

The authors have addressed all my comments. I recommend the publication of this manuscript.

Author Response

Reviewer 2

English language and style are fine/minor spell check required. The authors have addressed all my comments. I recommend the publication of this manuscript.

Reply: We would like to thank the Referee for carefully reading of our revised manuscript and raising valuable remarks.

The spell has been checked so that all spell mistakes is removed in the revised manuscript.  

Reviewer 3 Report

The authors propose a scheme to sense multiple parameters using a micro-ring resonator. A tunable laser is used to excite resonances of micro-ring and the presence of analytles changes the effective index, which is recorded in the transmittance depths of these resonant modes. Such dissipative sensing schemes have been elaborately studied in Ref. [20-26]. Here, the new feature is simultaneous detecting, with a single ring resonator, two distinct analytes which induce differential index changes in the two halves of the ring. Further, the recorded transmittance depths are processed using an artificial neural network and the performance is studied to demonstrate a reasonably low limit of detection.

I have the following comments which should be addressed:

The authors mention in the introduction that the usual reactive sensing schemes based on whispering gallery mode resonators like measuring the frequency shift/resonance broadening/mode splitting are expensive because they require a tunable laser source or a high resolution spectrometer. I agree, that the dissipative sensing, which requires only recording the transmittance on the other hand can be cheap. The introduction gives the impression that the authors want to use the dissipative sensing, instead of the reactive sensing, because it is a cheaper alternative. However, the proposed design requires a tunable laser source as it requires measurement of T for multiple resonances. In that, how is the proposed sensing scheme cheaper? Is it not true that frequency shift instead of measuring T can be processed by the ANN will be effectively same? These should be clarified in the introduction.

Also, very recently, sensing using a Bragg-grating imprinted on the ring and molecular attachment on parts of the ring have been proposed in J. Opt. 20 (2018) 085803 and Optics express 27 (24), 34997 (2019). There, the partial functionalization (similar to the proposed design, here) of the ring is also proposed. The presence of analyte is detected by minute splitting of the resonances, which can be excited by a broad band source and the beat signal can be recorded by conventional electronics. The self-referenced sensors do not require a tunable laser or a high-Resolution spectrometer and is cheap, while the low limit of detection is retained. These modern developments should be mentioned to avoid giving the impression that reactive sensing is always expensive. 

When two analytes are simultaneously sensed, the spectra (here, T) depend on the concentrations of both analytes C1 and C2. So, for every concentration C1 (in a specified range), C2 should be varied (in a specified range) and the transmission spectra needs be recorded. This increases the size of training set drastically. The situation becomes worse, when more than two analyte concentration needs to be measured. To demonstrate, the authors have assumed C1=C2. However, in real sample, C1 and C2 can be very different. This raises the concern about the feasibility of adopting the proposed design for sensing real samples.

Usually, bio-sensors envisioned to measure multiple analytes contain multiple ring resonators, each functionalized for a single analyte. For example, Anal. Chem. 82(1), 69–72 (2010), IEEE J. Sel. Top. Quantum Electron. 16(3), 654–661 (2010) and Analyst 141(18), 5358–5365 (2016). In the proposed strategy, for sensing two concentrations, if two rings (may of slightly different radii, so that the spectra do not overlap) are used instead, and one is functionalized for C1 and other for C2, the recorded spectra can of ring1 is independent of C2 and that of ring2 is independent of C1. Will that not lead to a smaller training set? The authors should justify, why is the adopted strategy (of using single rings) better than using two independent rings.

Also, another aspect to be care for is the relative position of the two functionalized region. As demonstrated in Nat. Nanotechnol. 6(7), 428–432 (2011) (and its Supplementary Information), the spectral response is a non-monotonic function of the relative position of particle binding. For using the proposed design as a commercial sensor, it might be challenging to functionalize exactly the same locations in all fabricated devices. For that purpose, it is important to know how does the proposed design tolerate changes in the relative positions of functionalization or is it immune to it?

The results seems feasible and can be of significance to the sensing community. I can recommend the article for publication, if the authors can provide clarifications for the above issues.

Author Response

Reviewer 3

The authors propose a scheme to sense multiple parameters using a micro-ring resonator. A tunable laser is used to excite resonances of micro-ring and the presence of analytles changes the effective index, which is recorded in the transmittance depths of these resonant modes. Such dissipative sensing schemes have been elaborately studied in Ref. [20-26]. Here, the new feature is simultaneous detecting, with a single ring resonator, two distinct analytes which induce differential index changes in the two halves of the ring. Further, the recorded transmittance depths are processed using an artificial neural network and the performance is studied to demonstrate a reasonably low limit of detection.

The results seems feasible and can be of significance to the sensing community. I can recommend the article for publication, if the authors can provide clarifications for the above issues.

Reply: First of all, we would like to thank the Referee for carefully reading of our revised manuscript and raising valuable remarks. In this manuscript, the multimode sensing mechanism results from the different responses to the analyte to be detected in multiple wavelength modes, i.e. the different transmission depth changes in the different resonant wavelengths. Inspired by multi-sensor information fusion technology, the main innovation point in the manuscript is that the multimode sensing makes full use of the effective sensing information in SIMRR, and provides a simple and universal approach to achieving the multi-parameter sensing. The numerical results indicate that our scheme of multi-parameter sensing in a multimode SIMRR holds great potential for practical high-sensitive sensing platforms comparing with the single-mode sensing based on WGM resonators.

I have the following comments which should be addressed:

1.The authors mention in the introduction that the usual reactive sensing schemes based on whispering gallery mode resonators like measuring the frequency shift/resonance broadening/mode splitting are expensive because they require a tunable laser source or a high resolution spectrometer. I agree, that the dissipative sensing, which requires only recording the transmittance on the other hand can be cheap. The introduction gives the impression that the authors want to use the dissipative sensing, instead of the reactive sensing, because it is a cheaper alternative. However, the proposed design requires a tunable laser source as it requires measurement of T for multiple resonances. In that, how is the proposed sensing scheme cheaper? Is it not true that frequency shift instead of measuring T can be processed by the ANN will be effectively same? These should be clarified in the introduction. 

Also, very recently, sensing using a Bragg-grating imprinted on the ring and molecular attachment on parts of the ring have been proposed in J. Opt. 20 (2018) 085803 and Optics express 27 (24), 34997 (2019). There, the partial functionalization (similar to the proposed design, here) of the ring is also proposed. The presence of analyte is detected by minute splitting of the resonances, which can be excited by a broad band source and the beat signal can be recorded by conventional electronics. The self-referenced sensors do not require a tunable laser or a high-Resolution spectrometer and is cheap, while the low limit of detection is retained. These modern developments should be mentioned to avoid giving the impression that reactive sensing is always expensive. 

Reply: Firstly, we are sorry that our writing is not clear enough to make the Referee think we are discussing which method is much cheaper in the manuscript. So, we rewrite the sentence in the lines 21-22 as “……the sensing is typically realized by measuring the spectral changes with high resolution using a tunable laser or an optical spectrometer……” in the revised paper. The two references (K.Cicek and Martin Cryan, Spectral analysis of a four-port DBR micro-ring resonator for spectral sensing applications, J. Opt. 20 (2018) 085803 and Nirmalendu Acharyya, Mohamed Maher, and Gregory Kozyreff, Portable microresonator-based label-free detector: monotonous resonance splitting with particle adsorption, Optics express 27 (24), 34997-35011 (2019).) has been added in the revised paper.

   Secondly, in the manuscript, the proposed multimode sensing requires a tunable laser source to obtain measurement of T for multiple resonances. However, it is necessary to emphasize the main innovation point in the manuscript. The main innovation point in the manuscript is that the multimode sensing makes full use of the effective sensing information in SIMRR, and provides a simple and universal approach to achieving the multi-parameter sensing. We deem that our works have particular novelty, because it solves the long-existing problem of detecting multiple parameters with a micro-ring in this field. The reasons are as follows:

 (b)

Fig.R1 (a) an add Drop Ring Resonator and (b) its typical amplitude transmittance (Fig.1(a) and Fig.2 (a) in Ref.[10])

In the conventional WGM sensors, the sensor has always realized by measuring relative spectral frequency shift of a single resonant mode. Fig.R1 shows an add Drop Ring Resonator and its typical amplitude transmittance (M. La Notte, B. Troia, T. Muciaccia, C. E. Campanella, F. De Leonardis and V. M. N. Passaro, “Recent advances in gas and chemical detection by vernier effect-based photonic sensors,” Sensors, 14, 4831-55 (2014).), and the peaks show a periodical oscillation in wavelength. When the WGM sensor is used for sensing, each resonant mode gives almost the same relative spectral frequency shift, and then the detection of a single resonant mode is enough for one-parameter sensing. Certainly, ANN is useless for such single mode sensing. But more importantly, it is impossible to realize the multi-parameter sensing using such single mode sensing, which is the long-existing problem of detecting multiple parameters in WGM sensors.

The multimode SIMRR sensing is firstly proposed in the paper. As shown in Fig.1, based on the dissipative sensing mechanism in specially designed self-interferenced microring resonator (SIMRR), the multimode sensing in a wide range of wavelengths has been proposed. The multimode sensing mechanism results from the different responses to the analyte to be detected in multiple wavelength modes, i.e. the different transmission depth changes in the different resonant wavelengths. Inspired by multi-sensor information fusion technology, the main innovation point in the manuscript is that the multimode sensing makes full use of the effective sensing information in SIMRR, and provides a simple and univeral approach to achieving the multi-parameter sensing.

Fig.1 Multimode sensing mechanism by SIMRR. (a) Structural diagram of SIMRR and its sensor design. (b) The simulated transmission spectra in the wavelength range of 1400-1600nm for the cases of the length change of sensing arm waveguide l=-10,0,10nm. (c) Enlarged diagram of dotted area in (b).

When two analytes are simultaneously sensed, the spectra (here, T) depend on the concentrations of both analytes C1 and C2. So, for every concentration C1 (in a specified range), C2 should be varied (in a specified range) and the transmission spectra needs be recorded. This increases the size of training set drastically. The situation becomes worse, when more than two analyte concentration needs to be measured. To demonstrate, the authors have assumed C1=C2. However, in real sample, C1 and C2 can be very different. This raises the concern about the feasibility of adopting the proposed design for sensing real samples.

Reply: It is most important to achieve the desired performance for the machine learning algorithm. It is not true that the large number of training set must be best for the proposed ANN, as shown in Fig.2(d) in the paper. If few number of training set is used to train the network, it is well when the network’s performance is good for test set. Indeed, in order to save the training time, the concentration C1 equals C2 when constructing the training set. However, in our test set, C1 and C2 is different, as shown in Fig 4.(c) and (d) (noted that Fig4.(c) is the comparison between the estimated values by BP-ANN and the theoretical value of gas concentration C1, and Fig4.(d) is the comparison between the estimated values by BP-ANN and the theoretical value of gas concentration C2, “The test set is selected when the gas concentration C1 is adjusted within the range of 0.050138% <C1<0.099848% with the step of 0.012426% and the gas concentration C2 is adjusted within the range of 0.052049%<C2<0.094905% with the step of 0.0071426%. Obviously, there are five values of gas concentration C1 and seven values of gas concentration C2 within a small concentration range of 0.05%<C1(C2)<0.1%. So the test set corresponds to 35 values of two gas concentrations C1 and C2, which are considered as the test indices.” The selection of test set almost cover the small concentration range, and the measurement does not lose its generality.

   Moreover, in another recent works of our group, a FP cavity is used to detect three gas concentrations by measuring its spectra with the help of ANN. In the example, the target parameters are three gas concentrations. The training set can be obtained when three gas concentrations are equal, and the theoretical and experimental works have demonstrated that ANN can achieve the desired performance for the test set. In addition, in the reference (Zahavy T, Dikopoltsev A, Moss D, et al, Deep learning reconstruction of ultrashort pulses, Optica, 2018, 5(5):666-673.), if it is difficult to obtain the training set in the experiment, it can be replaced with the training set of theoretical results.

For the perspective that the required data set increases when the number of analyte concentration increases, we agree with the Referee. This is a common problem in the machine learning, and we would spend more effort on this perspective in the future studies.

3.Usually, bio-sensors envisioned to measure multiple analytes contain multiple ring resonators, each functionalized for a single analyte. For example, Anal. Chem. 82(1), 69–72 (2010), IEEE J. Sel. Top. Quantum Electron. 16(3), 654–661 (2010) and Analyst 141(18), 5358–5365 (2016). In the proposed strategy, for sensing two concentrations, if two rings (may of slightly different radii, so that the spectra do not overlap) are used instead, and one is functionalized for C1 and other for C2, the recorded spectra can of ring1 is independent of C2 and that of ring2 is independent of C1. Will that not lead to a smaller training set? The authors should justify, why is the adopted strategy (of using single rings) better than using two independent rings.

Reply: Thanks for this insightful comments about the motivation of our work. The Referee provided very nice example of multiple ring resonators for measuring multiple analytes (Washburn, A. L., Luchansky, M. S., Bowman, A. L. & Bailey, R. C. Quantitative, label-free detection of five protein biomarkers using muliplexed arrays of silicon photonic microring resonators. Anal. Chem. 82, 69–72 (2010); Muzammil Iqbal, Martin A. Gleeson, Bradley Spaugh, Frank Tybor, William G. Gunn, Michael Hochberg,Tom Baehr-Jones, Ryan C. Bailey, and L. Cary Gunn, Label-free biosensor arrays based on silicon ring resonators and high-speed optical scanning instrumentation, IEEE J. Sel. Top. Quantum Electron. 16(3), 654–661 (2010); Washburn, Adam L. ; Shia, Winnie W. ; Lenkeit, Kimberly A. ; Lee, So-Hyun ; Bailey, Ryan C. Multiplexed cancer biomarker detection using chip-integrated silicon photonic sensor arrays, Analyst 141(18), 5358–5365 (2016)). In the revised manuscript, we have added the references.

   The main purpose in the manuscript is to achieve multiparameter sensing using a microring resonator, and it solves the long-existing problem of detecting multiple parameters with one microring in this field. Our proposed multimode sensing can realize the multi-parameter sensing without multiplexed arrays of silicon photonic microring resonators, thus is more compact.  On the hand, for many sensing scenarios, we cannot separate the influence of two different analytes or different parameters. For example, when measuring the environment refractive index, we can not distinguishing the temperature fluctuation and the changing of environment refractive index.

Therefore, comparing to the approach pointed out by the Referee, our method could be more compact in practical applications, is applicable to more general sensing scenario, and also be interesting for the fundamental studies of multiple parameter estimations. In the revised manuscript, we have also added one sentence to illustrate this point: “Although a micro-ring resonator biosensing array is exploited to realize multi-parameter biosensing by using wavelength division multiplexing technology, its complex array structure and detecting cost hinder its further development, ” and the separation of different parameters might not be available in practical sensing scenarios.

Also, another aspect to be care for is the relative position of the two functionalized region. As demonstrated in Nat. Nanotechnol. 6(7), 428–432 (2011) (and its Supplementary Information), the spectral response is a non-monotonic function of the relative position of particle binding. For using the proposed design as a commercial sensor, it might be challenging to functionalize exactly the same locations in all fabricated devices. For that purpose, it is important to know how does the proposed design tolerate changes in the relative positions of functionalization or is it immune to it?

Reply: Thanks for the nice comment about the design tolerance. In our proposal, the SIMRR sensor can be fabricated in Silicon-on-Insulator (SOI) technology with a standard SOI wafer characterized by a 2μm thick buried silica layer and a 220 nm thick silicon upper layer (M. La Notte, B. Troia, T. Muciaccia, C. E. Campanella, F. De Leonardis and V. M. N. Passaro, “Recent advances in gas and chemical detection by vernier effect-based photonic sensors,” Sensors, 14, 4831-55 (2014).). Specifically, the sensor chip can be fabricated partially by deep ultraviolet photolithography. The whole architecture is covered by depositing a 600 nm-thick SiO2 layer using plasma-enhanced chemical vapor deposition technology(M. La Notte, B. Troia, T. Muciaccia, C. E. Campanella, F. De Leonardis and V. M. N. Passaro, “Recent advances in gas and chemical detection by vernier effect-based photonic sensors,” Sensors, 14, 4831-55 (2014).). Next, the pattern of the sensing waveguide window is formed by using positive photoresist. Ultimately, the SiO2 upper-cladding as well as the SiO2 insulator layer beneath in the window region are removed by using HF wet etching, resulting in the suspended wire of the sensing arm waveguide. The SOI silicon waveguides with a single mode have a width of 500nm and a height of 220nm. For a two-parameter sensor, two functional sensitive material can be coated separately on the surfaces of two half-ring waveguides in two windows, and the operation can be performed in the laboratory. According to the current nanofabrication technology, we think it is not very challenge to dope the sensing material on different region of waveguide separately.

With the increasing of the number of target parameter, the number of functional sensitive material will be corresponding increase, and maybe it becomes difficult to decorate them in the corresponding windows. In our opinion, we believe that we can benefit from the further development of the nanofabrication technology. Additionally, the machine learning approach have certain tolerance of the structure imperfection or parameter uncertainty. Again, we thank the Referee for this insightful comment, we will further study the influence of the structure imperfection and parameter uncertainty in the future.

This manuscript is a resubmission of an earlier submission. The following is a list of the peer review reports and author responses from that submission.

Round 1

Reviewer 1 Report

The manuscript written by Hu et al describes a multi-parameter sensing method using machine learning. It is a theoretical work built upon the structure (self-interference micro-ring resonator) reported in their previous works. The manuscript is in general well-written and has good logic. It solves the long-existing problem of detecting multiple particles in this field. Therefore, I recommend the publication of this work with several minor comments.

In literature, detecting multiple particles simultaneously have been discussed before (see Physical Review A 83, 023803 (2011), Physical Review A 85, 063808 (2012), Physical Review A 85, 013801 (2012), and New Journal of Physics 15, 073030 (2013) for example). Although these works still focused on collective behaviors, I suggest the authors include these works into references. The characters in the figures are too small to read. Please consider enlarging them. Micro-resonators usually exhibit a considerable individual difference in terms of their transmission spectra. How large is this effect and how robust is the trained neuron network? Can we transplant the trained results from one resonator to another? Also, a similar question also refers to the situation when temperature fluctuations exist. The transmission spectrum used for the input of the neuron network is obtained using the transfer matrix method (Eq. (1)) which is phenomenologically correct. However, the spectra obtained from Eq. (1) is probably not quantitatively the same as the one obtained from FDTD approaches. How different are the spectra obtained through these two approaches?

Author Response

Reviewer 1

General Comments:

The manuscript written by Hu et al describes a multi-parameter sensing method using machine learning. It is a theoretical work built upon the structure (self-interference micro-ring resonator) reported in their previous works. The manuscript is in general well-written and has good logic. It solves the long-existing problem of detecting multiple particles in this field. Therefore, I recommend the publication of this work with several minor comments.

Reply: We also thank the Referee for carefully reading of our manuscript and raising valuable remarks. In the following, we address the Referee’s other comments one by one.

In literature, detecting multiple particles simultaneously have been discussed before (see Physical Review A 83, 023803 (2011), Physical Review A 85, 063808 (2012), Physical Review A 85, 013801 (2012), and New Journal of Physics 15, 073030 (2013) for example). Although these works still focused on collective behaviors, I suggest the authors include these works into references.

Reply: Thanks for bring these related references to our attention. These references have been added in the revised paper, and two additional references are also appended in the revised paper (X. Yi, Y.-F. Xiao, Y.-C. Liu, B.-B. Li, Y.-L. Chen, Y. Li, and Q. Gong, “Multiple-Rayleigh-scatterer-induced mode splitting in a high-Q whispering-gallery-mode microresonator”, Physical Review A 83, 023803 (2011); Y. Hu, L. Shao, S. Arnold, Y.-C. Liu, C.-Y. Ma, and Y.-F. Xiao, “Mode broadening induced by nanoparticles in an optical whispering-gallery microcavity”, Phys. Rev. A, 90, 043847 (2014).)

2. The characters in the figures are too small to read. Please consider enlarging them.

Reply: Thanks for the suggestion. We have enlarged all the characters in the figures in the revised manuscript, and make them more clear.

Micro-resonators usually exhibit a considerable individual difference in terms of their transmission spectra. How large is this effect and how robust is the trained neuron network? Can we transplant the trained results from one resonator to another? Also, a similar question also refers to the situation when temperature fluctuations exist.

Reply: Micro-resonators usually exhibit a considerable individual difference in terms of their transmission spectra, and indeed, every resonator should be trained individually, where the trained results can not be transplanted from one resonator to another. Technically speaking, two identical resonators cannot be fabricated currently, and then their spectra should exhibit the considerable difference. However, the train parameters (such as learning rate) and train algorithm should be applicable to all microresonators, regardless of the sample-to-sample variance.

We want to point out that our proposed sensing method can works well when temperature fluctuations exist. Temperature can also be adopted as one of sensing targets. If we consider its influence on the performance in the training step, and the detection can be immune from the temperature fluctuations in the test step, where the premise is that the similar temperature fluctuations should be present in the training and test step.   

The transmission spectrum used for the input of the neuron network is obtained using the transfer matrix method (Eq. (1)) which is phenomenologically correct. However, the spectra obtained from Eq. (1) is probably not quantitatively the same as the one obtained from FDTD approaches. How different are the spectra obtained through these two approaches? 

Reply: The transmission spectrum of SIMRR can be theoretically obtained by using transfer matrix method, and it can also be numerically obtained from FDTD approaches. There are some differences present between two obtained spectra, and they can be usually different from the spectra obtained experimentally due to the device fabrication error. In the paper, the multiparameter sensing can be proved based on the spectra obtained theoretically, and it can also be proved based on the spectra obtained numerically and experimentally. In the references (Tom Zahavy, Alex Dikopoltsev, Daniel Moss, Gil Ilan Haham, Oren Cohen, Shie Mannor, and Mordechai Segev, Deep learning reconstruction of ultrashort pulses [J], Optica, 2018,5(5):666-673), due to lack of experimental training data, the theoretical data can be used as training data and the well detection results can also be achieved.

During the submission of the revised manuscript, based on these theoretical results, we are trying to realize experimentally the multi-parameter scheme in multimode SIMRR by means of machine learning algorithms. We have found that the multimode sensing by means of machine learning has strong fault tolerance ability, and its results are little affected when an exception occurred experimentally for a group of training set data.

Reviewer 2 Report

The authors describe a theoretical study in which the transmission spectrum of a whispering gallery mode resonator coupled in two places to a feed line (a structure they call an SIMRR) is analysed using a neural network. In particular, the consequences of the absorption of an analyte onto receptors on the feedline and the WGM resonators are considered. They conclude that this method of analysis offers a route for a single WGM sensor to be used to sense multiple parameters.

As written, the main thrust of the paper is the use of neural networks to analyse WGM spectra - in particular the position and size of spectral lines. This approach has been described before; see for example

Vladimir A. Saetchnikov, Elina A. Tcherniavskaia, Gustav Schweiger, and Andreas Ostendorf "Classification of antibiotics by neural network analysis of optical resonance data of whispering gallery modes in dielectric microspheres", Proc. SPIE 8424, Nanophotonics IV, 84240Q (30 April 2012);
E. A. Tcherniavskaia and V. A. Saetchnikov APPLICATION OF NEURAL NETWORKS FOR CLASSIFICATIONOF BIOLOGICAL COMPOUNDS FROM THE CHARACTERISTICS OF WHISPERING-GALLERY-MODE OPTICAL RESONANCE, Journal of Applied Spectroscopy, Vol. 78, No. 3, July, 2011

The main portion of this paper is therefore not particularly novel. The section on the analysis of multiple analytes using receptors on the WGM resonator itself has more interest, and may be worth publishing in 'Sensors' after extensive reformatting. This is essentially just sections 4 and 5.

More specific remarks:

a) The authors describe this as a dissipative measurement technique, but their analysis is based on the change in optical length of the sensing arm or resonator section. This is therefore a dispersive shift. This is a significant discrepancy.

b) There is much irrelevant detail - for example hypothetical fabrication processes (line 77 onwards) are not necessary for a theoretical paper. Details of how neural networks work would be better omitted, as would the training data for different network sizes. (Incidentally, I suspect that the larger MSE error for bigger hidden layers might be because they typically take longer to train.)

c) Does this method present an advantage over direct homodyne detection of the transmitted signal through the sensing arm? An optical length change of ±10 nm is a phase shift of ±3 degrees - this could be readily detected.

d) The gas molar concentrations presented (which should have units throughout, it is unclear what is being defined here) are in the range 0.01 to 0.2. If these are really in molar concentration, ie mol per unit volume, then the levels analyzed are fatal for humans. My feeling is that this is not realistic.

e) The coupling constant k is set to 0.5 (line 94). This seems to mean to me that the resonator will in general be significantly overcoupled to the feed arm. If half the power is lost at each coupling section, the two sections will result in an external loss rate similar to the frequency of the resonator, and far greater than the resonator's internal losses.

Presentation points:

e) The manuscript would benefit strongly from being corrected by a fluent English speaker.

f) Fig 1 has text too small, no letters for panels, and no dotted area in (b).

g) Fig 2f - what are the three lines?

Author Response

Reviewer 2

General Comments:

The authors describe a theoretical study in which the transmission spectrum of a whispering gallery mode resonator coupled in two places to a feed line (a structure they call an SIMRR) is analysed using a neural network. In particular, the consequences of the absorption of an analyte onto receptors on the feedline and the WGM resonators are considered. They conclude that this method of analysis offers a route for a single WGM sensor to be used to sense multiple parameters.

As written, the main thrust of the paper is the use of neural networks to analyse WGM spectra - in particular the position and size of spectral lines. This approach has been described before; see for example

Vladimir A. Saetchnikov, Elina A. Tcherniavskaia, Gustav Schweiger, and Andreas Ostendorf "Classification of antibiotics by neural network analysis of optical resonance data of whispering gallery modes in dielectric microspheres", Proc. SPIE 8424, Nanophotonics IV, 84240Q (30 April 2012);
E. A. Tcherniavskaia and V. A. Saetchnikov APPLICATION OF NEURAL NETWORKS FOR CLASSIFICATIONOF BIOLOGICAL COMPOUNDS FROM THE CHARACTERISTICS OF WHISPERING-GALLERY-MODE OPTICAL RESONANCE, Journal of Applied Spectroscopy, Vol. 78, No. 3, July, 2011

The main portion of this paper is therefore not particularly novel. The section on the analysis of multiple analytes using receptors on the WGM resonator itself has more interest, and may be worth publishing in 'Sensors' after extensive reformatting. This is essentially just sections 4 and 5.

Reply: First of all, we also thank the Referee for carefully reading of our manuscript and raising valuable remarks. Although a machine learning algorithm has been used for biological agents and micro/nano particles classification based on WGM resonators (Vladimir A. Saetchnikov, Elina A. Tcherniavskaia, Gustav Schweiger, and Andreas Ostendorf "Classification of antibiotics by neural network analysis of optical resonance data of whispering gallery modes in dielectric microspheres", Proc. SPIE 8424, Nanophotonics IV, 84240Q (30 April 2012); E. A. Tcherniavskaia and V. A. Saetchnikov application of neural networks for classification of biological compounds from the characteristics of whispering-gallery-mode optical resonance, Journal of Applied Spectroscopy, Vol. 78, No. 3, July, 2011), we deem that our works have particular novelty, because it solves the long-existing problem of detecting multiple particles in this field. The reasons are as follows:

(b)

Fig.R1 (a) an add Drop Ring Resonator and (b) its typical amplitude transmittance (Fig.1(a) and Fig.2 (a) in Ref.[10])

Firstly, in abovementioned references, a simple classification method is carried out using a probabilistic neural network by connecting with the relative spectral frequency shift of a single resonant mode and the number of WGMs appearing within the free spectral range in WGM sensors. However, the classification method is unable to implement effective measurement; that is to say, for example, it cannot detect a gas concentration from a mixture of gases. The method can only solve “the classification problem” in machine learning, and cannot realize the fine measurement. Based a multimode SIMRR, our method using artificial neuron network can solve “the regression problem” in machine learning, and it can realize multi-parameter detection. (Two different types of Machine Learning: (1) regression problem: When the target variable that we’re trying to predict is continuous, we call the learning problem a regression problem. classification problem: When the output can take on only a small number of discrete values, we call it a classification problem.)

Secondly, the multimode SIMRR sensing is firstly proposed in the paper. In the conventional WGM sensors, the sensor has always realized by measuring relative spectral frequency shift of a single resonant mode. Fig.R1 shows an add Drop Ring Resonator and its typical amplitude transmittance (M. La Notte, B. Troia, T. Muciaccia, C. E. Campanella, F. De Leonardis and V. M. N. Passaro, “Recent advances in gas and chemical detection by vernier effect-based photonic sensors,” Sensors, 14, 4831-55 (2014).), and the peaks show a periodical oscillation in wavelength. When the WGM sensor is used for sensing, each resonant mode gives almost the same relative spectral frequency shift, and then the detection of a single resonant mode is enough for one-parameter sensing. However, it is impossible to realize the multi-parameter sensing, which is the long-existing problem of detecting multiple particles in WGM sensors. In this paper, as shown in Fig.1, based on the dissipative sensing mechanism in specially designed self-interferenced microring resonator (SIMRR), the multimode sensing in a wide range of wavelengths has been proposed. The multimode sensing mechanism results from the different responses to the analyte to be detected in multiple wavelength modes, i.e. the different transmission depth changes in the different resonant wavelengths. Inspired by multi-sensor information fusion technology, the main innovation point in the manuscript is that the multimode sensing makes full use of the effective sensing information in SIMRR, and provides a simple and univeral approach to achieving the multi-parameter sensing.

Fig.1 Multimode sensing mechanism by SIMRR. (a) Structural diagram of SIMRR and its sensor design. (b) The simulated transmission spectra in the wavelength range of 1400-1600nm for the cases of the length change of sensing arm waveguide l=-10,0,10nm. (c) Enlarged diagram of dotted area in (b).

Therefore, we deem that the contents in Section 2 and 3 are necessary and make that the paper has good logic.

More specific remarks:

a) The authors describe this as a dissipative measurement technique, but their analysis is based on the change in optical length of the sensing arm or resonator section. This is therefore a dispersive shift. This is a significant discrepancy.

Reply: We do not agree with the Referee. The change of sensing arm eventually leads to a dissipative response of resonances in the microring. In our proposed SIMRR, the dissipative sensing scheme is based on the transmission depth change when only the sensing arm waveguide or microring waveguide is exposed to the analytes to be detected. In our previous works (Ref.26, H. Ren, C.-L. Zou, J. Lu, Z.Le, Y.Qin, S.Guo, W.Hu, “Dissipative sensing with low detection limit in a self-interference microring resonator,” J. Opt. Soc. of Am. B , 36, 942-951 (2019).), the dissipative sensing has been explained in detail. As shown in Fig.R2 (b) (Ref.26, Fig.1(b)), the simulated transmission spectra near 1550 nm for the designed SIMRR are displayed as the induced optical path change d of sensing arm waveguide is increased from -10nm (far left) to 10nm (far right) with the step size of 0.5 nm . It is clear that the induced phase difference Δφ depends on d, which directly modify transmission spectra in both resonant wavelength and extinction. In the abovementioned sensing, the dissipative coupling occurs, and then the sensing is also named as dissipative sensing.

Fig.R2 (a) The total field of internal SIMRR includes the micro-ring mode field and sensing feedback-loop ring mode field, which are indicated by the dashed and dotted lines, respectively. (b) The simulated transmission spectra near 1550 nm when only d is changed from -10nm (far left) to 10nm (far right) with the step size of 0.5nm. (c) The simulated transmission spectra near 1550nm as the refractive index change  of entire SIMRR structure is changed from -1×10-5 RIU (far left) to 1×10-5 RIU (far right) with the step size of 1×1 0-6RIU. (Fig.1 in Ref.[26])

When the target is exposed to the entire SIMRR structure, for comparison, Fig. 1(c) shows the simulated transmission spectra near 1550 nm as the induced refractive index change  of the entire SIMRR structure. The phase difference Δφ induced by sensing arm waveguide does not change with the change of . Then, with the change of , it is clear that the transmission depth remains unchanged, and only its resonant wavelength is changed, so that only dispersive sensing can be realized by measuring the resonant wavelength shift.

b) There is much irrelevant detail - for example hypothetical fabrication processes (line 77 onwards) are not necessary for a theoretical paper. Details of how neural networks work would be better omitted, as would the training data for different network sizes. (Incidentally, I suspect that the larger MSE error for bigger hidden layers might be because they typically take longer to train.)

Reply: We thank the Referee for the careful reading of our work and the suggestions to help us improving the manuscript. The hypothetical fabrication processes have been deleted (line 77 onwards in the original manuscript). But the details of how neural networks work is still being preserved to make the readers to understand the method. Although the artificial neuron network has been used in many fields, the multi-parameter SIMRR sensing using ANN is novel method for many readers, more details and explanations will help the readers to better understand the method.

In addition, we have investigated our ANN codes carefully. When the number of hidden layers is large, the training time is enough in our program, where the enough iterations are set to control the operation of training step. In the artificial neuron networks, the error does not always descend with the increasing of the number of hidden layers. When the number of hidden layers is excessively large, the training time is long, and the network may fall in a trap of local minimization and the algorithms may over-fit the training set, which leads to the larger MSE error (E. Bahadir Prediction of prospective mathematics teacher’s academic success in entering graduate education by using back-propagation neural network[J]. Journal of Education and Training Studies, 2016,4(5):113-122).

c) Does this method present an advantage over direct homodyne detection of the transmitted signal through the sensing arm? An optical length change of ±10 nm is a phase shift of ±3 degrees - this could be readily detected.

Reply: In our original paper, the related questions may not be clearly explained. In Ref. [24-26], our previous theoretical and experimental works have amply proved that the dissipative single mode sensing or multimode sensing present an advantage over direct homodyne detection of the transmitted signal through the sensing arm, because the sensing based a resonator always have a higher sensitivity than the sensing based the corresponding waveguide. In the original paper, in Fig.1(b) and (c), the step size of 10nm is adopted, and the large step size is chosen in order that the spectral changes are easy to be identified in Fig.1 (b) and (c). With the small step size, the changes of transmission depths are not easily distinguishable in the figures, especially in Fig.1(b). In fact, the entire detection range is set from -10nm to 10nm. In Section 3, it is will be found that “and the number of units of 900 for training dataset are obtained by changing the length change l within the range of -9.99688nm ≤ l ≤ 9.99688nm with the step size of 0.02224nm”, where the step size is 0.02224nm, and even much smaller than this under intensity noises can also be detected. In Fig.5 (d), the multimode sensing should have the lower LOD than the single mode dissipative sensing, and this has also been confirmed in the present numerical simulation and theoretical calculations.

d) The gas molar concentrations presented (which should have units throughout, it is unclear what is being defined here) are in the range 0.01 to 0.2. If these are really in molar concentration, ie mol per unit volume, then the levels analyzed are fatal for humans. My feeling is that this is not realistic.

Reply: We thank the Referee for the careful reading of our work and the suggestions to help us improving the manuscript. Indeed, for two hybrid gases, the range of a molar concentration of 1% to 0.2 is not realistic. Therefore, in the revised paper, “Two gas concentrations are changed from a molar concentration of 0.05% to 0.1% with the step size of 0.0005%”. In the revised paper, in Fig.4 (e) (f) and Fig.6 (d), the corresponding results are re-calculated, the conclusions hasn't changed.

e) The coupling constant k is set to 0.5 (line 94). This seems to mean to me that the resonator will in general be significantly overcoupled to the feed arm. If half the power is lost at each coupling section, the two sections will result in an external loss rate similar to the frequency of the resonator, and far greater than the resonator's internal losses.

 Reply:

Fig.R3 (a) Near 1550nm, the transmission depth and resonant wavelength versus the optical path change for the cases of α =0.01, 0.05 and 0.1dB/cm. The dissipative and dispersive sensitivities versus key structural physical parameters, such as, (b) the waveguide loss coefficient α , (c) the power coupling coefficient k , (d) the micro-ring radius R , (e) the initial sensing arm waveguide length L0 , (f) dispersive sensitivity versus the waveguide loss coefficient as the target is exposed to the sensing arm waveguide or the entire SIMRR structure. (Fig.2 in Ref.[26])

As described SIMRR structure in Fig.R2 (a), the total field of internal SIMRR results from the interference interaction between the microring mode field and sensing feed-back-loop ring mode field. As shown in Fig.R3 (c), With the increase of k from 0 to 1, the dispersive sensitivity increases linearly, while the dissipative sensitivity reaches its maximum value at k=0.45. The SIMRR exhibits two different regimes known as weak coupling and strong coupling within the range of 0 ≤ k ≤ 1. Whether at the weak coupling region ( k is very little) or the strong coupling region ( k is close to 1), the micro-ring mode field or sensing feedback-loop ring mode field is weakened, which results in the decrease of the interference interaction between two mode fields. When the power coupling coefficient k is close to 0.5, their interference interaction is possibly strongest, and the largest dissipative sensitivity can be obtained.

   So in the paper, the directional coupling constant k is set to 0.5 in order to obtain relatively large change of transmission depths per unit of length change of sensing arm waveguide.

Presentation points:

e) The manuscript would benefit strongly from being corrected by a fluent English speaker.

Reply: We have revised the whole manuscript carefully to make its English more fluent.

f) Fig 1 has text too small, no letters for panels, and no dotted area in (b).

Reply: In Fig.1, all the characters in the figures have been enlarged, and the letters for panels and no dotted area in (b) have been added in the revised paper.

g) Fig 2f - what are the three lines?

Reply: In Fig.2(f), the theoretical values of length change are set within the range of  with the step of 0.4nm, 0.5nm and 0.8nm, respectively, and their estimated average values of length change after 500 simulations separately make up of three lines from right to left.

Round 2

Reviewer 2 Report

In order to understand this manuscript better, and in particular the transmission properties of the proposed structure, I decided to reproduce some of the plots, and found some worrying inconsistencies.

Fig 3b does not match Eqns 8 and 9. There is clearly a typo in Eqn 8 - the 5000 should be inside the exponential - but aside from this, both of these functions have a maximum of 0.025 * C = 0.00125, at \lamda = 1550. Therefore the maxima in Fig 3b should be at 2.45 + 0.00125 = 2.45125. But they are both at 2.4515.

I also plotted Eqn 10, and found that it did not match Fig 4b at all for k = 0.5, or indeed any value of k. Instead of narrow transmission dips, I found a mix of broad and narrow transmission peaks. This makes sense, at it is possible to set \lambda such that the denominator in Eqn 10 is small -> T is large. I also found peaks when I set k = 0, which is odd because then the ring should have no effect and the structure should simply behave as a transmission line.

Also, Fig 6 in the new manuscript is identical to that in the old manuscript, including old values of concentration - apart from panel d. C1 and C2 values of 0.01 (1%) should be outside the range of the new training data set (0.05% to 0.1%, line 254) but the performance, both with and without noise at a particular SNR is now better?

I am still unconvinced by the authors' handling of the parameter k. The introduction of weak and strong coupling regimes does not help. These terms are usually applied to coupling between two resonators. It would be better to talk about the ring being over- or under-coupled. Then it appears that the authors would like to aim for critical coupling. So far so good, and this is consistent with eg work on optimizing rf readout of semiconductor qubits. But the ring has a circumference of 2 \pi * 20 um = 200um = 0.02 cm. A loss of 0.1dBm /cm gives a loss of 0.002 dB and a internal Q of about 15000. This is reasonable, but implies a k value of 1/Q ~1e-4 for critical coupling.

There are too many inconsistencies in this manuscript for me to recommend publication.